

# Microbiome differences between river-dwelling and cave-adapted populations of the fish *Astyanax mexicanus* (De Filippi, 1853)

Patricia Ornelas-García[1], Silvia Pajares[2], Víctor M. Sosa-Jiménez[1], Sylvie Rétaux[3] and Ramsés A. Miranda-Gamboa[4]

[1] Departamento de Zoología, Instituto de Biología, Universidad Nacional Autónoma de México, Mexico City, Mexico

[2] Unidad Académica de Ecología y Biodiversidad Acuática, Instituto de Ciencias del Mar y Limnología, Universidad Nacional Autónoma de México, Mexico City, Mexico

[3] Paris-Saclay Institute of Neuroscience, Université Paris Sud, CNRS UMR9197, Université Paris-Saclay, Gif-sur-Yvette, France

[4] Instituto de Energías Renovables, Universidad Nacional Autónoma de México, Temixco, Morelos, Mexico

Corresponding author
Patricia Ornelas-García,
patricia.ornelas.g@ib.unam.mx

## ABSTRACT

Symbiotic relationships between host and microbiome can play a major role in local adaptation. Previous studies with freshwater organisms have shown that microbiome performs numerous important biochemical functions for the host, playing a key role in metabolism, physiology or health. Experimental studies in fish groups have found an effect of enzymatic activity of gut microbiota on a variety of metabolic processes. The goal of this study was to compare stomach microbiome from cave and surface *Astyanax mexicanus,* in order to evaluate the potential response of microbiota to contrasting environmental conditions and physiological adaptations of the host. Stomach microbiota was obtained from three different populations: Pachón cave, and two surface rivers (Rascón and Micos rivers). The stomach microbiome was analyzed using the Ion 16S Metagenomic kit considering seven variable regions: V2, V3, V4, V6-7, V8 and V9. A high diversity was observed across samples, including 16 phyla, 120 families and 178 genera. Gammaproteobacteria, Firmicutes, Bacteroidetes and Betaproteobacteria were the most abundant phyla across the samples. Although the relative abundance of the core OTUs at genus level were highly contrasting among populations, we did not recover differences in stomach microbiome between contrasting habitats (cave vs. surface rivers). Rather, we observed a consistent association between β-diversity and dissolved oxygen concentration in water. Therefore, and unexpectedly, the microbiota of *A. mexicanus* is not linked with the contrasting conditions of the habitat considered here but is related to water parameters.

## INTRODUCTION

Microbe-host associations are essential drivers of evolution and can play a role in adaptation and speciation. Across taxa, the microbiome is essential for regulating metabolism,

physiology, and health (*Brucker & Bordenstein, 2012*; *Franchini et al., 2014*; *Sevellec et al., 2014*). In fish groups for example, microbiota alter enzymatic activity in the gut to influence polysaccharides metabolism or fat storage (e.g., *Bäckhed et al., 2004*; *Hongoh, 2010*; *Nicholson et al., 2012*; *Sullam et al., 2012*; *Tetu et al., 2013*; *Franchini et al., 2014*). The microbiome-host evolution has been suggested as a 'hologenome' model, with the host microbiome viewed as an extension of its own genome, but with a faster evolution, and the possibility to exchange microorganisms (with their genes and associated functions) with the environment (*Shapira, 2016*).

Several functional relationships have been proposed between bacteria and their host: (1) commensal bacteria (*Cahill, 1990*), which can be neutral or beneficial to the host (*Prescott, Harley & Klein, 2005*); (2) symbiotic obligatory relationships, with a mutual benefit (*Perru, 2006*); (3) opportunistic bacteria, which could be pathogenic under certain circumstances; and (4) pathogenic bacteria, which are responsible for infectious diseases (*Falkow, 1997*). The knowledge of metagenomes can provide unique insights about how the relationship between microbial communities, environmental variation and evolutionary processes shapes organism's niche adaptation to particular habitats and can have a prominent role in the host speciation (*Franchini et al., 2014*; *Sevellec et al., 2014*; *Liu et al., 2016*; *Shapira, 2016*).

The technological advances in high throughput sequencing have given an unprecedented opportunity to comprehensively characterize bacterial communities (*Hugenholtz & Tyson, 2008*). Additionally, the methods for characterizing microbiome assemblages are changing rapidly, with a variety of technologies being used to assay DNA sequence variation. Among such methods, 16S rRNA gene amplicons is the method most commonly used to characterize bacterial community composition (*Kent et al., 2004*; *Nemergut et al., 2011*; *Poretsky et al., 2014*; *Liu et al., 2016*), particularly through the study of hypervariable V3 and V4 regions (*Yu & Morrison, 2004*; *Wang et al., 2007*). However, there is an ongoing debate regarding which region is the most convenient to get the best characterization of the microbiome diversity (*Zwart et al., 2002*; *Chakravorty et al., 2007*; *Youssef et al., 2009*; *Guo et al., 2013*; *Barb et al., 2016*). Previous studies have shown differences in diversity estimates among 16S rRNA regions (*Youssef et al., 2009*), while "universal" primers have shown preferential amplification across certain bacterial groups (*Ahantarig et al., 2013*; *Gofton et al., 2015*).

Populations inhabiting caves provide a unique opportunity to study how extreme environments can shape the evolution of organisms adapted to dark conditions. The troglobite populations of the fish genus *Astyanax*, which inhabit the karstic Sierra Madre Oriental in northeastern Mexico, are among the most studied cavefish groups because they represent an outstanding opportunity to understand local adaptation and trait evolution (*Jeffery, 2009*; *Gross, 2012*; *Yoshizawa et al., 2012*; *Gross et al., 2013*; *Hinaux et al., 2013*; *Keene, Yoshizawa & McGaugh, 2015*; *Ornelas-García & Pedraza-Lara, 2016*). From its discovery by Salvador Coronado in 1936, this emerging model organism has brought light into the study of the evolution of unique characteristics associated to living in dark conditions. The occurrence of its sister surface species inhabiting the same region allows the comparison of adaptations under contrasting environmental pressures (cave vs.

surface). Additionally, this model provides a unique opportunity to evaluate the role of the evolutionary history in the microbiome biodiversity.

Among the unique characteristics that distinguish hypogean systems are the absence of light and primary production, as well as stable annual temperature comparatively to the photosynthetically active epigean systems (*Culver & White, 2005*; *Tabin et al., 2018*). Hence, in comparison to their closely-related populations from surface habitat, *Astyanax* cavefish have evolved a series of adaptations to local conditions that could be considered as either regressive or constructive. For example, loss of eyes and pigments are considered as regressed characteristics, but they may confer a selective advantage for life in the dark (*Bilandžija et al., 2013*; *Moran, Softley & Warrant, 2015*), while an increase of the number and size of neuromasts (*Yoshizawa et al., 2012*), as well as augmented olfactory capabilities (*Protas et al., 2008*; *Bibliowicz et al., 2013*; *Hinaux et al., 2016*; *Blin et al., 2018*) are considered as constructive adaptations. Metabolic adaptations, such as higher body fat and blood glucose content, have also been described in *Astyanax* cave populations (*Riddle et al., 2018*; *Xiong et al., 2018*). This turn over of adaptive traits across alternative environments could also occur regarding inter-species associations, like those established between host and microbiome. A fundamental aspect in the study of such associations is to distinguish between two different groups of microbiota, one host-adapted core and one environmentally-determined group (*Shapira, 2016*).

The main goal of the present study was to compare the microbiome of cave and surface *A. mexicanus,* in order to evaluate the potential response of this microbiome to contrasting environmental conditions and physiological adaptations of the host. To this end, we sampled one cave and two surface populations in the wild that came from different river systems in order to (a) characterize microbiome structure across seven hypervariable regions of the 16S rRNA gene (Ion Torrent PGM$^{TM}$ 16S kit: V2, V3, V4, V6–V7, V8, and V9), and (b) identify relationships between microbial community structure and contrasting environments.

## MATERIALS AND METHODS

### Study area
We included one cavefish population (Pachón) from the Sierra de El Abra, a mountain range running north to south along the eastern margin of the Sierra Madre Oriental, a karstic region in northeastern Mexico (*Elliott, 2016*). El Abra is considered as one of the highest flowing karst springs worldwide (*Gary & Sharp, 2006*) and the presence of *A. mexicanus* cavefish populations have been reported in at least 30 caves in the region (*Elliott, 2016*; *Espinasa et al., 2018*). Additionally, we included two surface populations located at the vicinity of El Abra, between the Sierras Las Crucitas and Sierra de la Colmena (San Luis Potosí, Fig. 1). All locations form part of the Pánuco river basin, which drains to the Gulf of Mexico (*Hudson, 2003*).

### Sample collection and DNA extraction
Sampling was made from August 5th to 8th, 2016, during the wet season. A total of eight samples were collected in cave and surface habitats (Fig. 1): Pachón cave ($n = 3$,

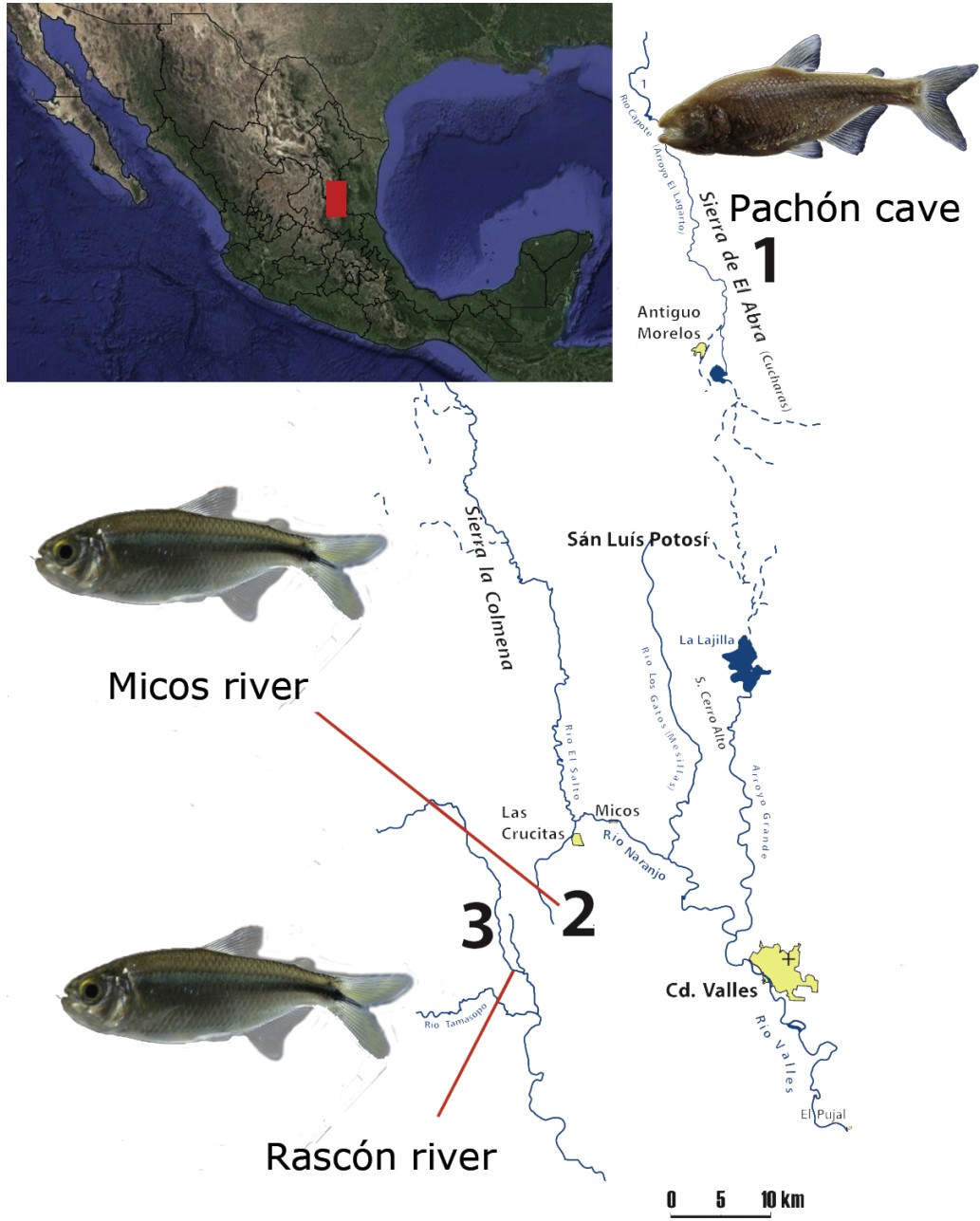

**Figure 1** Map of the sampling localities at the Sierra Madre Oriental (Mexico): 1) Pachón cave (El Abra region), 2) Micos river (at Otates locality) and 3) Rascón river (at Rascón locality). Map modified from *Elliott (2016)*.

two females and one male), Rascón river ($n = 2$, both males), and Micos river ($n = 3$, two females and one male). All the samples were collected under the auspices of the permit SGPA/DGVS/02438/16, delivered by SEMARNAT. Fish specimens were captured using hand nets, kept alive and rapidly transported in their environment water to a place with semi-sterile conditions to avoid environmental and cross-contamination. Fish were

euthanized following conditions stated in the collecting permit SGPA/DGVS/02438/16. The complete digestive tract was dissected and rinsed with sterile saline solution to remove any adherent material and preserved in 2 ml cryovials with Invitrogen™ RNAlater™ Stabilization Solution. The samples were kept in liquid nitrogen during sampling (5 days), and once in the laboratory were stored at −72 °C. All dissected specimens were preserved as vouchers in 95% ethanol and deposited in the Colección Nacional de Peces, IBUNAM (Mexico). Under sterile conditions and in a laminar flow hood, the stomach sack was excised from the rest of the intestine (approximately 1 cm$^2$) and immediately transferred into the lysing matrix provided by the FastDNA Isolation Kit for Soil (MP Bio, Santa Ana, CA, USA). The genomic DNA was extracted and purified following manufacturer's instructions. DNA quality and quantity were assessed by both, agarose gel electrophoresis and spectrophotometrically with a NanoDrop 2000 (Thermo Fisher Scientific, Waltham, MA, USA).

At each sampling point, seven physicochemical parameters were measured: pH, water temperature, dissolved oxygen (DO), electrical conductivity, total dissolved solids, salinity, and oxido-reduction potential (ORP), using a multiparameter Hanna HI9828 (HANNA Instruments).

## 16S rRNA library preparation and sequencing

The stomach microbiome for the eight *A. mexicanus* samples were assayed using the Ion 16S Metagenomic kit (LifeTech A2616), which uses seven hypervariable regions: V2, V3, V4, V6-7, V8 and V9. Amplification occurred in two multiplex pools (one for the V2, V4, and V8, and one for the V3, V6-7 and V9), with 25 PCR cycles, using the Ion Xpress Barcoded adapters. Amplicon sizes of the 6 fragments averaged 254 bp, within the range of 215–295 bp in *E. coli* (2014; Ion Toffent, Gilford, NH, USA). Libraries were quantified using Real-Time PCR, normalized and pooled. Emulsion PCR used Ion PGM Template OT2 400 kit. Sequencing was performed on an Ion Torrent PGM, using Ion 316 chip kit v2 BC with 4 and 8 chips.

## Sequences analysis

Bacterial taxonomic assignments were performed using Ion Reporter Metagenomics 16S software v5.2 (Life Technologies, Carlsbad, CA, USA) with the following analysis pipeline: (a) a minimum Phred score of Q20; (b) only reads with no mismatches in the barcode, up to three mismatches in the primer sequence, and a minimum length of 150 bp were included, and (c) reads were trimmed by primers at both ends. After primer trimming and length checking, reads were assigned to a hash table to get unique sequences and their abundance. Taxonomic assignment for unique sequences with a minimum of 10 copies was performed using MicroSEQ ID 16S Reference Library v2013.1 (Thermo Fisher Science) and Greengenes v13.5 (*McDonald et al., 2012*) databases. Reads were grouped into two bins: 90–97% match (approximating the family level) and >97–99% identity (genus level). This tool also provided information on relative taxonomic abundance by consensus and by primer within the dataset.

Rarefaction curves and alpha diversity indices were calculated using the alpha diversity script in the Quantitative Insights into Microbial Ecology software (QIIME v. 1.9.0;

*Caporaso et al., 2010*). For the diversity estimates, sequences were rarefied at 88,076 reads, which was the smallest read count across all fish samples (sample Micos female 1). The species richness was estimated with the Chao index, and diversity was determined by Shannon index (that quantifies the uncertainty in predicting the species identity in a random sampling) and Simpson index (that quantifies evenness and dominance of the species in the sample) (*Hill et al., 2003*).

The number of unique and shared OTUs (Operational Taxonomic Units) at genus level between sexes and populations was visualized with Venn diagrams using the Euleer package in R software (*R Core Team, 2015*). We evaluated the capability of each sequenced region to describe the stomach microbiota composition through the survey of the relative proportion of bacterial taxonomic groups retrieved by each hypervariable region of the 16S rRNA gene. Posteriorly, a non-metric multidimensional scaling (NMDS) ordination plot using Bray–Curtis dissimilarity matrix was constructed with the vegan package in R to depict community structure patterns in two dimensions. Permutation tests ($n = 1,000$) were used to fit significant water physico-chemical properties ($P < 0.1$) onto the ordination plot as vectors. Analysis of similarity (ANOSIM) was used to test for significant differences in microbiome structure between fish samples (999 permutations, $P < 0.001$). To analyze the distribution of OTUs at family level with relative abundances >1% of the total reads across all the samples, a heatmap was constructed using the Manhattan dissimilarity matrix and Ward's hierarchical clustering algorithms with the plots package in R. The 16S rRNA sequences were deposited in the NCBI Sequence Read Archive (SRA) under the Bioproject PRJNA487659 (SAMN09907737–SAMN09907742).

## RESULTS

### Diversity and composition of bacterial community in wild populations obtained by 16S rRNA variable regions

A total of 1,969,556 reads were recovered, with an average length of ∼200 bp. We observed that all rarefaction curves from fish samples attain an asymptote (Fig. S1), indicating that the sequencing effort was sufficient to cover the bacterial community. Alpha diversity indices varied among the fish samples, but didn't show a clear pattern among populations (Table 1). In terms of species richness, samples from Pachón cave showed the highest Chao index values, followed by Micos and finally Rascón. Diversity indices across samples were similar, with a range for the Shannon diversity index between 2.26 and 3.65, while for the Simpson diversity index the samples varied between 0.614 and 0.884, with the lowest values in both indices for the Rascón male 1. Finally, sex did not seem to impact biodiversity indexes (Fig. S2).

In total, we recovered 16 phyla, 120 families, and 178 genera, where the four following lineages accounted for 93.8% of total sequences across all the samples: Gammaproteobacteria (31.7%), Firmicutes (29.9%), Bacteroidetes (16.2%) and Betaproteobacteria (15%). Differences in composition and distribution of phylogenetic groups across populations were recovered (Fig. 2). The stomach microbiome of surface Rascón samples was dominated by Bacteroidetes (61.4% male 1 and 29.5% male 2) and

**Table 1** Alpha diversity indices of 16S rDNA gene sequences (at 97% of similarity) for the fish samples.

| Samples | No. reads | Observed genera | Chao1 | Shannon | Simpson |
|---|---|---|---|---|---|
| Rascon ♂1 | 336,118 | 30 | 30 | 2.26 | 0.614 |
| Rascon ♂2 | 353,478 | 53 | 53 | 3.64 | 0.884 |
| Micos ♀1 | 273,178 | 30 | 30 | 3.13 | 0.809 |
| Micos ♀2 | 299,903 | 55 | 55 | 3.30 | 0.825 |
| Pachon ♂1 | 329,915 | 65 | 65 | 3.65 | 0.864 |
| Pachon ♀2 | 384,847 | 64 | 64 | 3.58 | 0.874 |

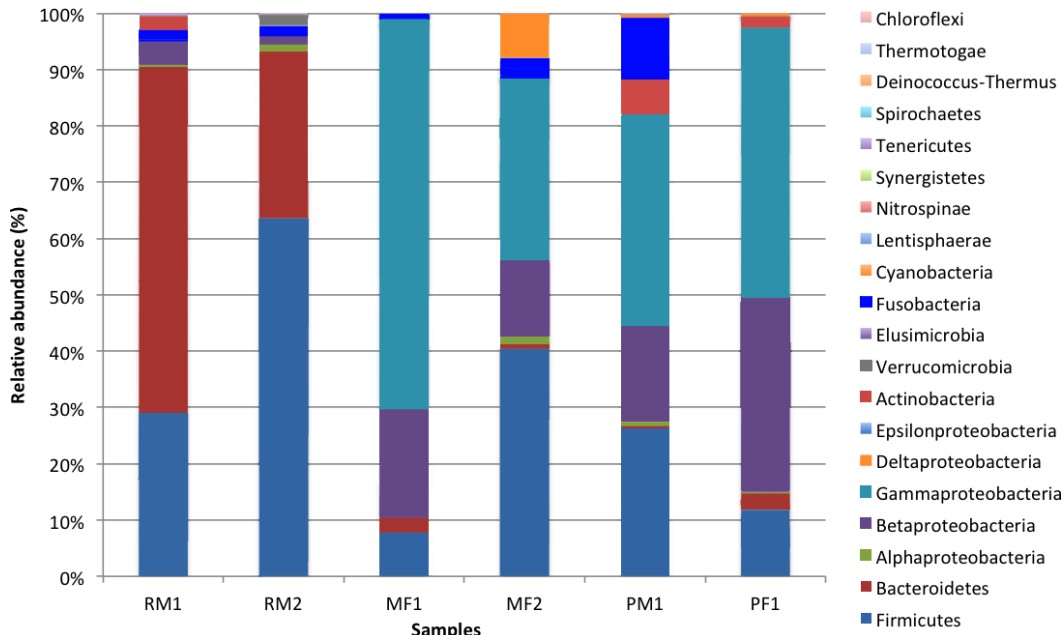

**Figure 2** Relative abundance of bacterial taxonomic groups (at the level of phylum, and class for the Proteobacteria) from the fish samples. Symbol abbreviations: R, Rascón; M, Micos; P, Pachón; M, Male; F, female.

Firmicutes (29% male 1 and 63.5% male 2), with a lesser proportion of Actinobacteria, Verrucomicrobia and Fusofacteria. In contrast, a different microbiota was recovered from Pachón cave samples, which were dominated by Gammaproteobacteria (37.6% male 1 and 47.9% female 1), Betaproteobacteria (17% male 1, and 34.5% female 1) and Firmicutes (26.3% male 1 and 11.9% female 2). The microbiome from Micos surface population was more similar to the Pachón cave population than to the Rascón surface population, being Gammaproteobacteria (68.5% female 1 and 32.2% female 2), Betaproteobacteria (19.4% female 1 and 13.6% female 2) and Firmicutes (7.8% female 1 and 40.5% female 2) the most abundant groups, with a lesser proportion of other groups such as Cyanobacteria and Fusobacteria.

## Comparison of 16S rRNA hypervariable regions for assignment of microbial diversity

The percentage of mapped reads was not homogenous among the hypervariable regions (Fig. S3). The region with the highest percentage of mapped reads was V3 (from 31.8% in female 1 to 44.9% in female 2, both from Micos river), followed by V8 (from 9.5% in Micos female 2 to 29.6% in Micos female 1), V4 (from 11.7% in Rascón male 2 to 18% in Rascón male 1 and Pachón female 1), and V6–V7 (from 6.6% in Pachón female 1 to 20.9% in Rascón male 1). The regions with the lowest percentage of mapped reads were V2 (from 1.4% in Rascón male 2 to 10.9% in Pachón female 1) and V9 (from 0.8% in Rascón male 1 to 12% in Micos female 1). In addition, not all the regions could distinguish the different taxonomic groups in the same way. For instance, V3 was the region with the greatest coverage of taxonomic groups (from six to nine groups), allowing the detection of the four most abundant groups (Gammaproteobacteria, Firmicutes, Betaproteobacteria, and Bacteroidetes) across all the fish samples. On the other hand, V9 covered the least number of taxonomic groups (from 1 to 2 groups) and could only detect Gammaproteobacteria and Betaproteobacteria groups.

## Shared and dominant stomach microbiome between *A. mexicanus* populations

The core stomach microbiome shared by the three *A. mexicanus* populations was formed by eight genus OTUs (Fig. 3) belonging to *Acinetobacter, Aeromonas, Citrobacter* and *Klebsiella* (class Gammaproteobacteria), *Prevotella* (Bacteroidetes), *Vogesella* (class Betaproteobacteria), *Clostridium* (Firmicutes), and *Cetobacterium* (Fusobacteria). The abundance distribution of the core microbiome differs among the three populations: Rascón river had the highest abundance of *Prevotella* (89.9%), while Micos river had more *Acinetobacter* (60.6%) and *Vogesella* (11.8%), and Pachón cave had more *Vogesella* (40.4%) and *Acinetobacter* (34.3%). Although they come from contrasting environments, Micos river and Pachón cave specimens shared more OTUs between them, than with samples from Rascón river. In addition, the most abundant bacterial groups per locality showed a clear difference across populations (Table 2): the dominant genus OTUs in Pachón cave samples belonged to Proteobacteria, Firmicutes and Actinobacteria, in Micos river samples belonged to Proteobacteria and Firmicutes, and in Rascón river samples belonged to Proteobacteria, Firmicutes and Bacteroidetes.

## Environmental determinants for the microbiome composition across populations

As the contrast between environments (cave vs. surface) did not seem to account for the observed differences in stomach microbiota, we assessed the association of water physicochemical variables with respect to β-diversity by means of NMDS analysis (Fig. 4), which grouped the samples in two clusters: bacterial community from the Rascón river samples formed a unique cluster, and those from Micos river and Pachón cave formed another cluster. This strong clustering trend was also confirmed by the ANOSIM ($p < 0.001$) analysis. Fitting the physicochemical variables onto the ordination plot showed that dissolved oxygen (DO) was the only significant explanatory variable for the bacterial

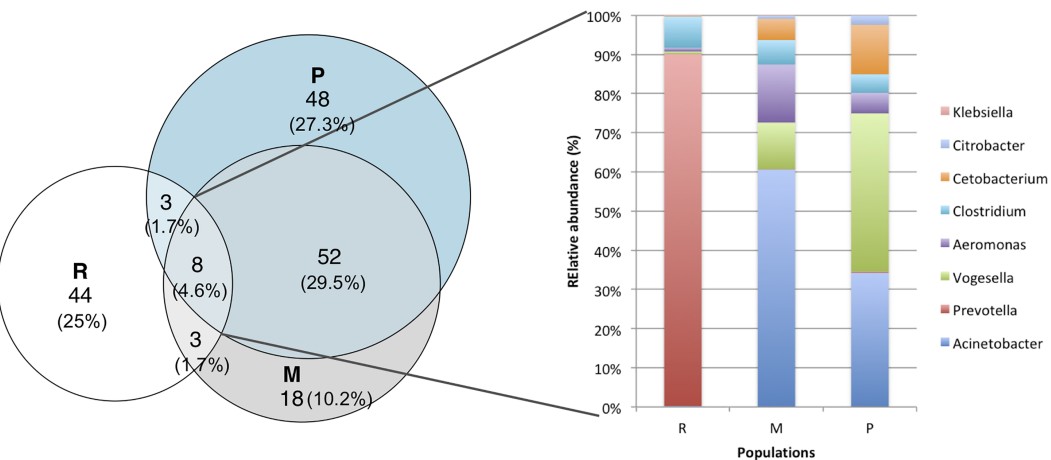

**Figure 3  Venn diagrams showing the numbers of shared and unique genus OTUs (at 97% of similarity) among populations.** Also shows the relative abundance of bacterial genera shared among the three populations (core microbiome). Symbol abbreviations: R, Rascón; M, Micos; P, Pachón.

**Table 2  Relative abundance of dominant OTUs at genus level expressed as mean values of the percentage of total bacteria.**

| Population | OTUs (Genus level) | Abundance (Typical % of total bacteria) |
|---|---|---|
| Pachón | *Dolosigranulum* (F) | 27.6 |
| | *Mycobacterium* (A) | 22.3 |
| | *Providencia* (P) | 17.8 |
| | *Enterococcus* (F) | 12.7 |
| Micos | *Sarcina* (F) | 21.3 |
| | *Legionella* (P) | 16.6 |
| | *Neorickettsia* (P) | 13.8 |
| | *Limnohabitans* (P) | 8.1 |
| Rascón | *Faecalibacterium* (F) | 13.3 |
| | *Rosevuria* (F) | 12.9 |
| | *Blattabacterium* (B) | 10.9 |
| | *Sutterella* (P) | 9.9 |

**Notes.**
F, Firmicutes; P, Proteobacteria; A, Actinobacteria; B, Bacteroidetes.

community structure. The DO values were considerably lower in both Pachón cave (2.97 mg/L) and Micos river (4.43 mg/L) relative to that from Rascón river (8.2 mg/L). Interestingly, observed differences in other variables such as conductivity, for which contrasting values were obtained (e.g., 226 µS/cm and 580 µS/cm for Rascón and Micos rivers, respectively), did not emerge as important factor as DO did for stomach microbiota community distribution (Table 3).

The heatmap of family OTUs with relative abundance >1% of the total reads showed the same pattern of grouping as the NMDS analysis (Fig. 5), separating the stomach
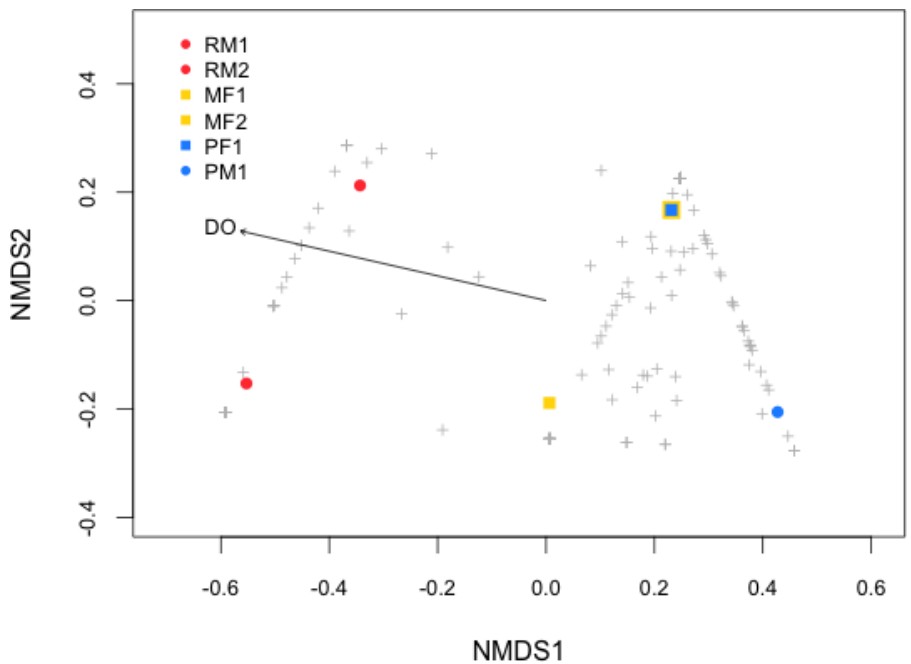

**Figure 4** **NMDS (Non-metric Multidimensional Scaling) plot of the samples (see symbols) based on Bray–Curtis distance matrix of genes OTUs from fish stomach microbiome.** Physicochemical properties were fitted onto the NMDS ordination and only significant variables are shown ($p < 0.1$). Symbol abbreviations: RM1, R, Rascón; M, Micos; P, Pachón; M, male; F, female.

**Table 3** **Values of physicochemical parameters for the three localities and statistical analysis ($p$-values) of the non-metric multidimensional scaling (NMDS) for the two ordination scores.**

|  | Rascón | Micos | Pachón | NMDS1 | NMDS2 | r2 | Pr(>r) |
|---|---|---|---|---|---|---|---|
| pH | 8.2 | 8.19 | 7.34 | −0.710 | −0.703 | 0.481 | 0.230 |
| Temperature (°C) | 26.75 | 26.24 | 25.09 | −0.867 | −0.497 | 0.696 | 0.070 |
| Conductivity (μS/cm) | 580 | 226 | 478 | −0.667 | 0.745 | 0.458 | 0.281 |
| Salinity | 0.28 | 0.11 | 0.23 | −0.672 | 0.740 | 0.463 | 0.273 |
| Dissolved oxygen (mg/L) | 8.2 | 4.41 | 2.97 | −0.991 | −0.133 | 0.961 | 0.010** |
| Total dissolved solid (ppt) | 290 | 113 | 239 | −0.667 | 0.745 | 0.458 | 0.281 |

**Notes.**
** Significance codes: **, 0.05.
$P$ values based on 1,000 permutations.

microbiome in two groups: the first one included the samples from Rascón river, and the second one included a mixture of both surface and cave environments (Micos river and Pachón cave). Moraxellaceae (Gammaproteobacteria) was the most abundant family with 14.5% of the total reads, followed by Enterobacteriaceae (Gammaproteobacteria) with 11.5% of the total reads and Prevotellaceae (Bacteroidetes) with 11.1% of the total reads. In general, fish samples from Micos river and Pachón cave showed greater abundance of bacteria belonging to Moraxellaceae and Enterobacteriaceae, while fish samples from Rascón showed greater abundance of bacteria belonging to Prevotellaceae.
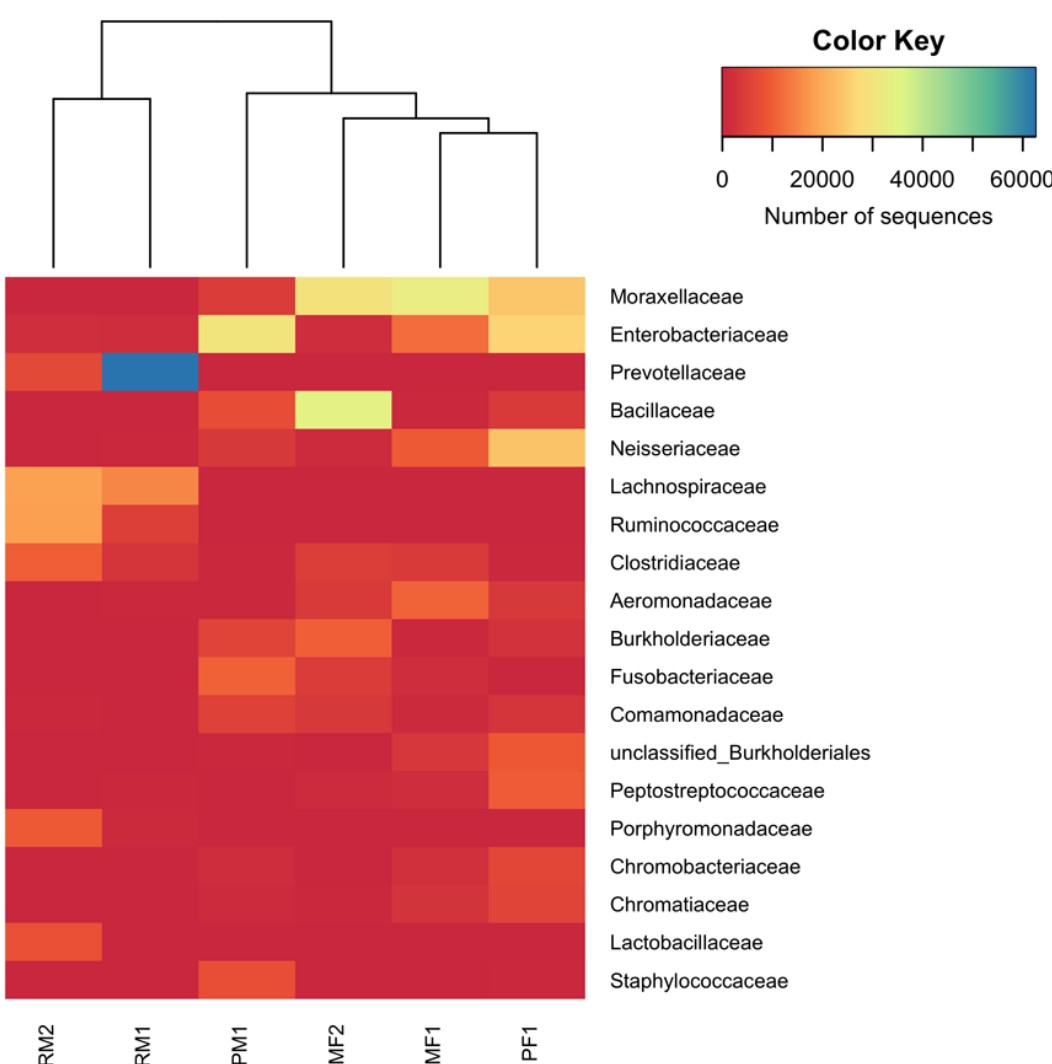

**Figure 5** **Heatmap of family OTUs with relative abundances > 1% of the total reads.** The Manhattan dissimilarity matrix and Ward's hierarchical clustering algorithms were applied in this analysis. The color key scale represents the number of sequences detected for each fish sample from lower (red) to higher (blue) abundance. Rows indicate family OTUs and columns indicate the different fish samples. Symbol abbreviations: R, Rascón; M, Micos; P, Pachón; M, male; F, female.

# DISCUSSION

## Different hypervariable regions of the 16S rRNA show a differential bacterial identification of the stomach microbiome of *A. mexicanus*

Next generation sequencing techniques have revolutionized the way of studying microbial diversity. However, inherent biases arise due to insufficient coverage of primers (*Hong et al., 2009*; *Ghyselinck et al., 2013*), primer-template mismatches (*Sipos et al., 2007*), and unequal amplification (*Polz & Cavanaugh, 1998*), distorting the comprehensive view of complex bacterial communities. The choice of the primer sets and covered regions can deeply affect the profile of microbiome composition (*Kumar et al., 2011*; *Soergel et al., 2012*). Therefore,

the use of multiple primers targeted to amplify different hypervariable regions of the 16S rRNA gene is an innovative strategy in metagenomic protocols that allows detecting a larger array of bacterial groups and enabling a more accurate microbiome description (*Milani et al., 2013*; *Barb et al., 2016*).

Previous studies in fish microbiota only targeted one or two hypervariable regions (typically V1–V3, V4, V6 or V8) (*Rawls et al., 2006*; *Kim, Brunt & Austin, 2007*; *Sun et al., 2009*; *Sullam et al., 2012*; *Franchini et al., 2014*; *Liu et al., 2016*). Here, we reported the first fish microbiota characterization incorporating the sequencing of seven hypervariable regions of the 16S rRNA gene (V2, V3, V4, V6-7, V8, and V9) using the Ion 16S $^{TM}$ Metagenomics Kit. This methodology has been recently reported for other models (*Barb et al., 2016*; *Cornejo-Granados et al., 2017*; *Sperling et al., 2017*), suggesting that more than one hypervariable region is needed to provide reliable diversity inferences (*Barb et al., 2016*).

In this study, the V3, V4 and V8 regions were the most accurate both for number of assigned reads per sample and for the OTUs distribution, while the other regions seem to underestimate the bacterial diversity. Among the hypervariable regions of the 16S rRNA gene, V3 and V4 are the less biased and the most used for microbiome studies (*Cai et al., 2013*; *Milani et al., 2013*). Additionally, studies using the Ion 16S$^{TM}$ Metagenomics Kit showed differences in the microbiome diversity depending on the region analyzed, recovering similarly to our results that the regions V3, V4 and V8 were the most informative-less biased regions (*Cornejo-Granados et al., 2017*; *Sperling et al., 2017*). Our results in the stomach microbiome characterization of *A. mexicanus*, as previous studies, showed the benefits of using different non-overlapping regions, and the potential biases of single marker identification in the microbiome characterization, which could affect not only the richness but also diversity in microbiome studies.

### Environmental factors influencing microbiota diversity of A. mexicanus

Vertebrate microbiome is a complex microbial ecosystem containing diverse and abundant bacteria, archaea and fungi, whose structure can be influenced by several factors, such as host genetics, type of environment, trophic level, fish density, physicochemical conditions and seasonality (*Sullam et al., 2012*; *Liu et al., 2016*).

In this context, we compared stomach microbiome from surface and cave-dwelling *A. mexicanus*, to evaluate the potential response of stomach microbiota under contrasting environmental conditions. Unexpectedly, the results show that patterns of microbial diversity are more related to local environmental conditions rather than to the contrasting nature of habitats (cave vs. surface). That is, water DO concentration seems to play a major role in the stomach microbiome diversity.

Emerging research suggests that the core microbiota in *A. mexicanus* is related to environmental water and sediment bacteria, represented by Bacteroidetes, Proteobacteria, Firmicutes and Actinobacteria (*Zwart et al., 2002*; *Austin et al., 2012*; *Liu et al., 2016*), which are abundant in microbiota of other vertebrates as well (*Franchini et al., 2014*; *Sevellec et al., 2014*; *Liu et al., 2016*; *Xiong et al., 2018*). This is interesting because these taxonomic groups are shared across a wide diversity of marine and freshwater fish, suggesting the

existence of a core microbiota shared between a broad range of species (*Franchini et al., 2014*; *Sevellec et al., 2014*; *Liu et al., 2016*).

Previous studies have suggested a correlation between fish microbiome and evolutionary history (*Ley, Peterson & Gordon, 2006*), diet (*Liu et al., 2016*), and/or environment (e.g., salinity *Sullam et al., 2012*). Thus, even when environment can affect the ecology and population dynamics of microorganisms (*Babich, Stotzky & Ehrlich, 1980*), the effect of abiotic factors remains understudied in fishes' microbiota (*Sullam et al., 2012*). We did not find evidence that contrasting conditions such as those represented by cave and surface environments have a significant effect on the stomach microbiota diversity of *A. mexicanus*, suggesting that other factors could be the main determinants in the bacterial consortia in this study system. Previous studies have reported the lack of differentiation of microbiota of fish species inhabiting different habitats (freshwater vs. saltwater), finding a significant interaction between trophic level and habitat conditions with the gut microbiome diversity (*Sullam et al., 2012*). Previous studies have also found evidence that water quality, temperature, salinity and seasonality are among the abiotic conditions affecting fish microbiome diversity (*Cahill, 1990*).

### Commensals and pathogens in A. mexicanus microbiome

Even if major differences in stomach microbiota structure could be explained by DO concentration, differences in richness and diversity across fish samples may reflect some symbiotic interactions between the microbiome communities and nutrients assimilation from host's diet, as reported in previous studies (e.g., *Liu et al., 2016*). It was suggested that cellulose-degrading bacteria such as *Citrobacter, Leptotrichia, Bacillus* and *Enterobacter* are the dominant groups in herbivorous fish, while *Clostridium* has been reported at higher abundances in omnivorous and filter-feeding fishes. Protease-producing bacteria, such as *Cetobacterium* and *Halomonas*, have been reported in fish with piscivorous habits (*Liu et al., 2016*). In this study, fish samples from Micos river and Pachón cave showed a larger abundance of bacteria belonging to *Vogesella, Acinetobacter, Cetobacterium* and *Citrobacter*, tentatively associated to cellulose-degrading metabolism. Similar microbiome composition was reported for omnivorous fish (*Liu et al., 2016*).

In the fish samples from Pachón cave *Citrobacter* was the OTU with the largest relative abundance, which seems counter-intuitive regarding the lightless and vegetal-free environment of their hosts. However, *Citrobacter* could play a role in polysaccharides metabolism, since cellulose digestion in fish depends on exogenous cellulose activity (i.e., diet-dependent), suggesting stomach microbiota could contribute to their digestion (*Saha & Ray, 1998*). Additionally, as a part of the core microbiome in fish samples from Pachón cave, we observed the presence of *Klebsiella* genus, which is associated to tanniase metabolism (tannin acyl hydrolase). Tannin is a substance found in different plants groups (*Ray, Ghosh & Ringø, 2012*). In this regard, vegetal material could be introduced to the Pachón cave during the rainy season, corresponding to a resource for the fish nutrition, and which digestion could be microbiome-dependent. Further analyses of the carbon and nitrogen metabolism could shed more light about the trophic levels of these populations. Contrastingly, *Prevotella* was the most abundant genus in the Rascón river population.

Previous studies have reported *Prevotella* in the intestinal mucus of the rainbow trout *Oncorhynchus mykiss* (*Kim, Brunt & Austin, 2007*), and it has been linked to plant-rich diets, as well as with chronic inflammatory conditions in other fish species (*Ley, Peterson & Gordon, 2006*).

In general terms, the microbiome diversity found in *Astyanax* surface and cave-adapted fish was similar to that previously reported for another omnivorous fish (*Liu et al., 2016*), in which stomach microbiome contributes to several enzymatic activities related to polysaccharides digestion. In this respect, considering the omnivorous trophic habits of a closely-related species, *Astyanax aeneus*, which has been shown to display a high capability to exploit a wide variety of trophic resources (*Ornelas-García et al., 2018*), it is possible that a similar trophic strategy is depicted by surface *A. mexicanus*. Several studies have documented trophic habits of the troglobitic populations of *A. mexicanus*, with evidence of detritivore habits (*Wilkens & Burns, 1972*; *Hüppop, 1986*). By contrast, a recent study evaluated in detail the food regime in individuals from the Pachón cave, and found an ontogenetic change in trophic habits. During post-larvae stage *Astyanax* fish are active predators of water fleas (Cladocera), copepods, ostracods, isopods and other insects. In contrast, in the adult stage, the stomach content was dominated by detritus (possibly bat guano), and preys as complete flies and beetles, which tentatively were ingested alive (*Espinasa et al., 2017*). Further studies are required including additional cave populations in order to better explore the diversity in stomach microbiome across caves and surface populations, and its relevance for the host metabolism.

Finally, we recovered some groups of bacteria that have been reported as pathogenic or opportunistic, which can cause a disease once the fish is exposed to stressful conditions. Among the exclusive OTUs within Pachón cave population, we recovered the genera *Mycobacterium*, *Rothia* (both Actinobacteria), and *Providencia* (Proteobacteria), all of them reported as pathogenic bacteria in freshwater fish (*Austin et al., 2012*). Additionally, *Acinetobacter*, *Aeromonas* and *Cetobacterium*, have been reported as dominant microbiota in the lake whitefish (*Coregonus clupeaformis*) and several other species, in which a pathogenic role has been suggested (*Sevellec et al., 2014* and references within). Further studies are required to better understand the contrasting dynamics of the microbiome between surface and cave-dwelling *Astyanax* fish, as well as those opportunistic bacterial groups.

## CONCLUSIONS

*Astyanax mexicanus* stomach microbiome resembles that observed in other fish groups, and further that reported for other vertebrates, and is composed of Bacteroidetes, Proteobacteria, Firmicutes and Actinobacteria. Although we did not recover differences in stomach microbiome between contrasting habitats (caves vs. surface), the relative abundance of the core OTUs at genus level was highly contrasting among populations. We observed a consistent association between stomach β-diversity and water DO concentration. Therefore, microbiota in *A. mexicanus* could result from a combination between physicochemical water conditions and its diet. Further investigation including other cave fish populations, and focus on seasonal variation, as well as diet characterization,

could shed light to our understanding of the microbiome-host symbiotic interactions under contrasting environments.

## ACKNOWLEDGEMENTS

We would like to thank Luis Espinasa, Carlos Pedraza Lara, Victor Simon, Ulises Rivera and Angeles Verde, who helped with specimen collection for the current study. We also want to thank to MR Riddle for her kind suggestions on an early version of the Manuscript. Laura Margarita Márquez Valdelamar and Patricia Rosas Escobar, Sequencing Service, Instituto de Biología, UNAM (IBUNAM), for their help with the Library construction of 16S rRNA amplicons for the Metagenomic Ion kit. Pachón cavefish picture was taken by Carmen Loyola, IBUNAM.

### Funding

This work was supported by UNAM-PAPIIT, project IA203017 and by the Ecos-Nord CONACYT exchange program grant No. 279100. Ramsés Miranda Gamboa received a Grad student Fellowship at the Posgrado en Ciencia e Ingeniería de Materiales, Universidad Nacional Autónoma de México. Víctor Sosa Jiménez received a Grad student Fellowship at the Posgrado en Ciencias Biológicas, Universidad Nacional Autónoma de México. The funders had no role in study design, data collection and analysis, decision to publish, or preparation of the manuscript.

### Grant Disclosures

The following grant information was disclosed by the authors:
UNAM-PAPIIT: IA203017.
Ecos-Nord CONACYT exchange program: 279100.
Posgrado en Ciencia e Ingeniería de Materiales, Universidad Nacional Autónoma de México.
Posgrado en Ciencias Biológicas, Universidad Nacional Autónoma de México.

### Competing Interests

The authors declare there are no competing interests.

### Author Contributions

- Patricia Ornelas-García conceived and designed the experiments, performed the experiments, analyzed the data, contributed reagents/materials/analysis tools, prepared figures and/or tables, authored or reviewed drafts of the paper, approved the final draft.
- Silvia Pajares conceived and designed the experiments, analyzed the data, contributed reagents/materials/analysis tools, prepared figures and/or tables, authored or reviewed drafts of the paper, approved the final draft.
- Víctor M. Sosa-Jiménez and Ramsés A. Miranda-Gamboa performed the experiments, approved the final draft.
- Sylvie Rétaux authored or reviewed drafts of the paper, approved the final draft.

## Animal Ethics

The following information was supplied relating to ethical approvals (i.e., approving body and any reference numbers):

SEMARNAT: SGPA/DGVS/02438/16.

## Field Study Permissions

The following information was supplied relating to field study approvals (i.e., approving body and any reference numbers):

Field collection permit was approved by the Secretaria del Medio Ambiente y Recursos Naturales (SEMARNAT: SGPA/DGVS/02438/16).

## Data Availability

Sequence Read Archive, Bioproject PRJNA487659 (SAMN09907737–SAMN09907742).

## Supplemental Information

Supplemental information for this article can be found online at http://dx.doi.org/10.7717/peerj.5906#supplemental-information.

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
