# Peer review of "Microbiome differences between river-dwelling and cave-adapted populations of the fish Astyanax mexicanus (De Filippi, 1853)"

_PeerJ, doi:10.7717/peerj.5906_

## Round 0.1 · original submission · Minor Revisions

I have now obtained two reviews of your paper and both reviewers found it very interesting and well done. They both, however, have a number of issues for you to take care of before acceptance. They detail these issues in their respective reviews. I think it will be fairly straightforward for you to accommodate these concerns. Good luck with your revision.

·

Basic reporting

This manuscript is written in a clear professional English with a good logical-flow. Literature reference and background information are sufficient and adequate. The manuscript structure, figs and tables meet requirement of proficiency, and Raw sequence data was shared. Authors provided clear hypothesis "cavefish and surface fish have distinct gut microbiota" and their result did not support. I suggested slightly adjusted hypothesis that their result supports.

Experimental design

The experimental design meets the aim and scope of PeerJ: The good quality of original research in Biological Sciences. Research question is clear and well defined. I found a potential major issue that authors did not enphasize how they chose stomach as the tissue they studied. The midgut and hind-gut are more popular to investigate the gut microbiota composition. Authors need to provide clear explanation for it. Nonetheless, the result filled the gap of the knowledge of the gut microbiota compositions in the wild surface fish and cavefish. The study meets high technical and ethical standard, and the described method is sufficient to replicate by others.

Validity of the findings

Authors' finding is surprising to me because there is no difference in between gut biota compositions of surface and cave populations. Instead, there was a difference between populations living in different oxygen concentrations. Authors suggested further investigations in other cave populations to enrich the full-view of the symbiotic evolution between the gut microbiota and cavefish. I personally would like to see larger sample size per population (N ~ 8) rather than N= 2~3 in this study. However, I understand that there is a high hurdle to sample wild-cavefish because these populations are endangered.
The most of data and analysis seems robust. I left comments where I wondered such as Fig. 5: In this figure, Prevotellaceae is extremely high in RM1 individual. I would like to know there is no technical error (such as a bug in their script) in the analysis. The conclusion follows what they found, and is linked with original questions. Authors did not make any unreasonable speculations.

Additional comments

The followings are line-by-line comments

L25: “Stomach biopsies…”
I assume that authors meant stomach microbiota. Biopsy is confusing because authors euthanized fish and dissected out the entire digestive tract. Biopsy is, in general, taking a sample tissue from a living organism. I first thought that authors recovered the stomach microbiota by using a thick syringe needle in the field.

L33: “beta-diversity”
It is difficult for non-ecologists to recognize what beta-diversity is. You may state “stomach microbiota composition” or else instead of beta-diversity. However this is my personal opinion.


Introduction:
Authors wrote a quite good summary to inform about the gut microbiota. Can authors use the term “commensal” in the Discussion section starting from L328- “Symbionts and pathogens in A.mex microbiome”? Authors made a good job to discuss about opportunistic/pathogenic bacteria in the later of this discussion section. The first half of this section seems for commensal bacteria. It will be clearer for readers if you declare that you start talking about commensal bacteria.

L71 reference “(Jeffery 2009; Gross 2012;…)
As a suggestion, you may add “(Keene et al., 2015)” Keene AC, Yoshizawa M, McGaugh SE: Biology and Evolution of the Mexican Cavefish. Amsterdam: Elsevier Inc.; 2015.

L79-L82: this sentence is too long so that confusing. You may split it into two sentences.

L88: references for olfactory capabilities, you may add:
(1) Blin M, Tine E, Meister L, Elipot Y, Bibliowicz J, Espinasa L, Rétaux S (2018) Developmental evolution and developmental plasticity of the olfactory epithelium and olfactory skills in Mexican cavefish. Dev Biol in press. Available at: https://doi.org/10.1016/j.ydbio.2018.04.019.
(2) Protas M, Tabansky I, Conrad M, Gross JB, Vidal O, Tabin CJ, Borowsky R (2008) Multi-trait evolution in a cave fish, Astyanax mexicanus. Evol Dev 10:196–209.

L95-L97: “Differences between surface and cave-dwelling populations…”
This sentence describes authors' initial aim. However, the result does not support it. Although PeerJ can accept negative result, to impress readers positively, I suggest authors to restate. For example, “Determining the degree of the contribution of biotic and abiotic factors that are associated with the stomach biota composition, in the present study, we compare the [stomach] microbiome of cave and surface A. mexicanus, in order to…”


Method:
Sample size is substantially low (N=2~3 per pop) by taking account of the high variability of gut microbiota, which can change between different ages, different diets, and sick vs. healthy states in human (Luzupone et al 2012***). However, I understand that it is difficult to gather large sample sizes for endangered populations. I believe that authors’ data is still informative for cave animal societies.
I have very limited knowledge for Ion Torrent system, thus, cannot evaluate the validity of how author processed. Besides of it, checking V2-V9 regions of rRNA genes is solid and valid, ANISOM analysis seems valid too.
L173: typo in “OTUS”, it should be “OTUs”.
L177: Please provide the submission number for SRA

*** Lozupone CA, Stombaugh JI, Gordon JI, Jansson JK, Knight R (2012) Diversity, stability and resilience of the human gut microbiota. Nature 489:220–230.


Results:
There is no major issue in Result. It was well written and easy to follow the logical-flow.

L187 and L189: Please provide brief explanations for “Chao index” and “Simpson index” for non-ecologists.

L248: Just double check if “226 S/cm” and “580 S/cm” are still right digits. I think these are extremely high. For example, Rohner et al. 2013 reported that Nacimento Del Rio Choy was 1324 µS, which is 0.001324 S/cm.

Discussion:
L291-L293: It may be good to emphasize A. mexicanus. For example, “using different amplicon regions is beneficial to the studies of the A. mexicanus microbiome too.”

In the section of "Environmental factors influencing microbiota diversity of A. mexicanus."
Please discuss a possibility that cave and surface populations eat different diets that influences their gut microbiota. It is critical to refer this. You may refer [Espinasa L, Bonaroti N, Wong J, Pottin K, Queinnec E, Rétaux S (2017) Contrasting feeding habits of post-larval and adult Astyanax cavefish. Subterr Biol 21:1–17.]

Fig 1.
Lost a red line from Pachon fish. Provide explanations what blue and yellow shades represent (lakes and cities?), and what hatched and solid blue lines stand for.

Fig. 2.
Color codes are difficult to follow. You may make them solid colors instead of gradient colors, and draw arrows between the index color boxes and bar graphs. You may draw arrows only the major taxonomic goups including Firmicutes, Beta-proteobacteria, gamma-proteobacteria, and Actinobacteria.

Fig. 3.
Please show the relative abundance of bacterial genera for 52 that shared between M and P, also show 44 that is unique for R. these graphs inform the candidates that are sensitive for oxygen concentration.

Fig 5
Please double check if Prevotellaceae is truly accumulated in RM1 rather than calculation-mistake. This odd score or Prevotellaceae might skew the conclusion although authors wisely chose rank-test for most of the stats.

Reviewer 2 ·

Basic reporting

No comment.

Experimental design

The Introduction could use some improvement on framing the study in light of current knowledge of fish microbiomes and vertebrate gut microbiomes in general. There are some more recent reviews on both that can be found in Shapira (2016); Colston & Jackson (2016) and I recommend Vatsos 2016 for experimental design considerations specific to fish microbiome studies.

Additionally the authors need to address how differences in diet and environment between these populations (particularly cave vs surface) could influence their findings. Omnivorous fish tend to have gut microbiomes similar to free-living communities in the water column whereas that isn't necessarily the case for strict carnivores. In some cases gut microbiome isn't lined to ecotype or diet (Sullam et al. 2015). It is unfortunate that the authors did not sequence environmental samples as a control and comparison, as this extremely limits the ability to draw conclusions from their results.

The methodology regarding sample collection needs to be expanded. As written it is unclear whether fish were collected and stored in native water or water that was sterilized prior to sample collection or the conditions under which dissections were made to ensure sterility. The authors do not specify the amount or type of gut material used in extractions. After dissections were samples stored at room temperature or flash frozen? All of these minutia must be detailed in microbiome studies both for purposes of evaluating methodology and for replication.

Validity of the findings

I can only comment that the rigor of the analyses are sound but without clarification on collection methods and controls in sampling design there is not enough information to verify the validity of the author's claims.

Additional comments

This is a very interesting and important study. I do think that results could be biased by comparing 2x the number of surface samples and populations to a single cave population. I appreciate that the authors have used multiple primer sets to evaluate the efficacy of a single set to yield comprehensive data. However, there is a wealth of information on fish microbiomes (reviewed in Colston & Jackson 2016) to which the authors could add their data for a meta-analyses type study on this topic, too.

Finally, when making concluding statements such as "...resembles that observed in other fish groups..." etc the authors need to specify the taxonomic level to which they refer. In the broadest sense 16S surveys of vertebrate guts suggest all vertebrate gut microbiomes "resemble" one another so this statement carries little weight.

---

## Round 0.2 · accepted · Accept

Thank you for your careful revisions and attention to the previous reviewers' critiques. I feel your paper is an exciting addition to the journal and is now ready for me to recommend acceptance.

#